# Analytical Approximations of Well Function by Solving the Governing Differential Equation Representing Unsteady Groundwater Flow in a Confined Aquifer

**Manotosh Kumbhakar** *,† and **Vijay P. Singh**

Department of Biological and Agricultural Engineering, Texas A&M University, College Station, TX 77843, USA
* Correspondence: manotosh.kumbhakar@gmail.com
† Current address: Department of Civil Engineering, National Taiwan University, Taipei 10617, Taiwan.

**Abstract:** A solution of the governing equation representing the drawdown in a horizontal confined aquifer, where groundwater flow is unsteady, was first provided by Theis and is famously known as the *Theis solution*. This solution was given in terms of an exponential integral, also called the *well function*, for which simple and reliable approximations are preferred due to their practical applications. To that end, several approximations are available in the literature, of which some are based on series approximations for the integral, and others are numerical approximations. This study employs three kinds of homotopy-based methods, namely the homotopy analysis method (HAM), homotopy perturbation method (HPM), and optimal homotopy asymptotic method (OHAM), for analytically solving the governing partial differential equation (PDE). For convenience, the PDE is first converted to a boundary value problem (BVP) using a similarity transformation. Comparing the series approximations obtained using these methods with the Theis solution, it is found that the 10th-order HAM, and just three terms of OHAM-based solutions, provide accurate approximations. On the other hand, the HPM-based solution is found to be accurate only within a small domain. Further, the proposed approximations are compared with several series and numerical approximations available in the literature using the percentage error. The proposed methodology provides accurate approximations of the well function by directly solving the governing differential equation in a general framework and thus can be adapted to other practical situations arising in groundwater flow.

**Keywords:** *Theis solution*; *well function*; homotopy analysis method; asymptotic series expansion; confined aquifer

**MSC:** 76-10

## 1. Introduction

Pumping test analysis in groundwater flow is one of the key methods for determining aquifer parameters, such as transmissivity and storativity. Unsteady groundwater flow in confined aquifers is described by coupling the continuity equation and the Darcy flux law. The resulting equation is a partial differential equation (PDE) of diffusion type. Under simplified assumptions, approximate analytical solutions of the PDE have been derived. One common assumption is that the confined aquifer is of uniform thickness and is homogeneous and isotropic. Using Boltzmann transformation, Theis [1] transformed the partial differential equation to an ordinary differential equation and then derived a solution in terms of an exponential integral, which came to be known as the well function, and gave tabular values of this function for different values of the Boltzmann variable or the argument of the well function. Theis found the unsteady flow of groundwater to be analogous to the unsteady flow of heat in a homogeneous solid and derived the desired solution.

In general, the exponential integral can occur in many applications of transient groundwater flow, hydrological problems, mathematical physics, and applied mathematics ([1–3]). The direct explicit evaluation of this integral may not be analytically tractable. Thus, most of the evaluations are based on approximations using either series expansion or numerical behavior. In groundwater studies, this integral is commonly known as the well function. Expressing this function as a power series and considering only two terms of the series, an approximate solution has also been proposed [4]. However, this series approximation is valid only within a small domain. The asymptotic (divergent) series can also be proposed for large values of the argument [5]. Many studies that are valid within a restricted domain are available for finding the approximation, based on polynomial or rational approximation or series expansions ([6–11]). Swamee and Ojha [12] combined several approximations valid for a specific region of the argument to provide an approximation to the well function. Barry et al. [13] constructed an approximation using the interpolation between large and small asymptotes. In recent work, Vatankhah [14] proposed a simple and very accurate approximation for the well function by combining the approximations of small and large values of the argument. However, these studies are based on the computational aspects of the exponential integral, not the solution methodology of the governing equations.

Differential equations play one of the most important roles in modelling in science and engineering. Therefore, the methods for solving these equations analytically are key topics of research. Most of the mathematical tools available are valid only for a specific class of nonlinear equations. On the other hand, Liao [15] proposed a methodology, namely the homotopy analysis method (HAM), using the basic concept of homotopy from topology to solve nonlinear equations analytically in series form. It was shown that the method provided great freedom for solving nonlinear problems, and its applicability was not confined to a small class of problems [16,17]. Following the usefulness of the method, two other variants, namely, the homotopy perturbation method (HPM) and optimal homotopy asymptotic method (OHAM), have also been proposed [18,19]. One of the key advantages of these methods is that they do not depend on the presence of a small/large parameter in the governing equation or boundary conditions. While these methods have been applied in several disciplines of science and engineering, their application to water engineering problems is limited. Further, apart from the well-known series expressions, the approximation of the well function using the closed-form formula mentioned in the previous paragraph is based on empirical observation or the characteristics of the function. Thus, the change in flow configuration or incorporation of other flow factors, which changes the solution to the problem, cannot be explained by the proposed approaches. Following this, the objective of this work is to apply the homotopy-based methods to unsteady groundwater flow in a confined aquifer and obtain an approximation for the well function by solving the governing differential equation itself. For that purpose, first, the methods are briefly described in a general framework and then applied to the considered problem. The convergence of the series solution is also discussed.

## 2. Brief Overview of Homotopy-Based Methods

For the convenience of the reader, here, we describe a general framework for three variants of homotopy-based methods. It may be noted that all of these methods are based on the fundamental concept of homotopy from topology, which describes the continuous deformations between two mathematical objects. Specifically, two objects can be called homotopic if one can be continuously deformed into the other. The circle and square in 2D and the doughnut and coffee cup in 3D are two examples of homotopy. In the context of differential equations, this idea was extended by [15], as they represent curves (or functions) in a mathematical sense. Liao [15] developed the homotopy analysis method (HAM), which was further extended by deriving two variants of this method, namely the homotopy perturbation method (HPM) and the optimal homotopy asymptotic method (OHAM). These three methods are employed in this paper.

### 2.1. Homotopy Analysis Method

Let us write a general differential equation in the following form:

$$\mathcal{N}(u(x,t)) = 0 \tag{1}$$

where $\mathcal{N}$ is the nonlinear operator or the original operator of the equation; $u$ is the dependent (unknown) variable; and $x$ and $t$ are the independent variables, e.g., space and time. Now, the zeroth-order deformation equation is constructed as follows [16]:

$$(1-q)\mathcal{L}[\Phi(x,t;q) - u_0(x,t)] = q\hbar H(x,t)\mathcal{N}[\Phi(x,t;q)] \tag{2}$$

subject to the boundary conditions:

$$\mathcal{B}\left(\Phi, \frac{\partial\Phi}{\partial x}, \frac{\partial\Phi}{\partial t}\right) = 0 \tag{3}$$

Here, $q$ is the embedding parameter; $\Phi(x,t;q)$ is the representation of solution across $q$; $u_0(x,t)$ is the initial approximation; $\hbar$ is the auxiliary parameter; $H(x,t)$ is the auxiliary function; and $\mathcal{L}$ and $\mathcal{N}$ are the linear and nonlinear operators, respectively. Following Equation (2), the core idea of HAM is that as $q$ varies from 0 to 1, $\Phi(x,t;q)$ transforms from the initial approximation to the final solution. Mathematically, at $q = 0$, $\Phi(x,t;0) = u_0(x,t)$, and at $q = 1$, $\Phi(x,t;1) = \mathrm{u}(x,t)$. Now, the higher-order terms need to be determined. For that purpose, the following higher-order deformation equation is used [16]:

$$\mathcal{L}[u_m(x,t) - \chi_m u_{m-1}(x,t)] = \hbar H(x,t) R_m\left(\vec{u}_{m-1}\right), \ m = 1,2,3,\dots \tag{4}$$

where

$$\chi_m = \begin{cases} 0 \text{ when } m = 1, \\ 1 \text{ otherwise} \end{cases} \tag{5}$$

and

$$R_m\left(\vec{u}_{m-1}\right) = \frac{1}{(m-1)!} \left.\frac{\partial^{m-1}\mathcal{N}[\Phi(x,t;q)]}{\partial q^{m-1}}\right|_{q=0} \tag{6}$$

where $u_m$ for $m \geq 1$ are the higher-order terms. The derivation of Equation (4) requires the successive differentiation of the zeroth-order deformation Equation (2). The final solution can now be obtained as follows:

$$u(x,t) = u_0(x,t) + \sum_{m=1}^{\infty} u_m(x,t) \tag{7}$$

For the assessment of the solution, Equation (7) is truncated in order to have an approximate solution. In the framework of HAM, several operators and functions need to be chosen to obtain the solution. Liao [16] proposed some fundamental rules, namely the rule of solution expression, the rule of coefficient ergodicity, and the rule of solution existence, which will be discussed in the next section.

### 2.2. Homotopy Perturbation Method

Let us rewrite the differential equation as follows:

$$\mathcal{N}(u(x,t)) = f(x,t) \tag{8}$$

Now, the homotopy that satisfies [18] is constructed:

$$(1-q)[\mathcal{L}(\Phi(x,t;q)) - \mathcal{L}(u_0(x,t))] + q[\mathcal{N}[\Phi(x,t;q)] - f(x,t)] = 0 \tag{9}$$

where the symbols carry the same meaning as mentioned in the previous section. Additionally, similar to HAM, Equation (9) shows that at $q = 0$, $\Phi(x,t;0) = u_0(x,t)$, and at $q = 1$, $\Phi(x,t;1) = u(x,t)$. Let us now express $\Phi(x,t;q)$ as a series in terms of $q$, as follows:

$$\Phi(x,t;q) = \Phi_0 + q\Phi_1 + q^2\Phi_2 + q^3\Phi_3 + q^4\Phi_4 + \dots \tag{10}$$

where $\Phi_m$ for $m \geq 1$ are the higher-order terms. As $q \to 1$, Equation (10) produces the final solution as:

$$u(x,t) = \lim_{q \to 1}\Phi(x,t;q) = \sum_{k=0}^{\infty} \Phi_k \tag{11}$$

First, the series Equation (10) is substituted into Equation (9), and then the like powers of $q$ are equated to obtain the HPM-based solution, Equation (11). In comparison with HAM, the HPM does not contain any additional auxiliary function and auxiliary parameters, which restricts its applicability and also the rate and region of convergence of the series [16,17]. Indeed, the HPM is a special case of HAM, subject to the same set of linear and nonlinear operators and unit auxiliary function when the auxiliary parameter $\hbar = -1$.

### 2.3. Optimal Homotopy Asymptotic Method

In some cases, HPM and HAM require several terms of the series solution in order to obtain a good approximation. Therefore, the optimal homotopy asymptotic method (OHAM) was developed with the aim of obtaining an accurate solution with just two to three terms of the series. To that end, Marinca and Herisanu [19] proposed OHAM by using asymptotic expansions of the functions and operators, which is described below. We consider the differential equation in the following form:

$$\mathcal{L}[u(x,t)] + \mathcal{N}[u(x,t)] + h(x,t) = 0 \tag{12}$$

subject to the boundary conditions:

$$\mathcal{B}\left(u, \frac{\partial u}{\partial x}, \frac{\partial u}{\partial t}\right) = 0 \tag{13}$$

where symbols denote the same variables as discussed in the previous section. Following HAM, one can construct the homotopy as:

$$(1-q)\left[\mathcal{L}(\Phi(x,t,C_j;q)) + h(x,t)\right] = H(x,t,C_j;q)\left[\mathcal{L}(\Phi(x,t,C_j;q)) + h(x,t) + \mathcal{N}(\Phi(x,t,C_j;q))\right] \tag{14}$$

where symbols have their usual meaning, and $C_j$ are the unknown parameters that need to be determined. The auxiliary function is defined as:

$$H(x,t,C_j;q) = 0 \text{ for } q = 0 \neq 0 \text{ for } q \in (0,1] \tag{15}$$

It can be verified from Equation (14) that at $q = 0$, $\Phi(x,t,C_j;q) = u_0(x,t)$, and at $q = 1$, $\Phi(x,t,C_j;q) = u(x,t)$, which is the same as HAM and HPM, i.e., as $q$ goes from 0 to 1, we have the continuous deformation from the initial approximation to the final solution. The initial approximation $u_0(x,t)$ should be determined by solving the following equation:

$$\mathcal{L}(u_0(x,t)) + h(x,t) = 0 \tag{16}$$

subject to the boundary conditions:

$$\mathcal{B}\left(u_0, \frac{\partial u_0}{\partial x}, \frac{\partial u_0}{\partial t}\right) = 0 \tag{17}$$

Equation (16) can be constructed after setting $q = 0$ in Equation (14). In the framework of OHAM, one of the key steps is to expand the auxiliary function in terms of $q$, as follows:

$$H(x,t,C_j;q) = qH_1(x,t,C_j) + q^2 H_2(x,t,C_j) + q^3 H_3(x,t,C_j) + \ldots \tag{18}$$

where $H_i(x,t,C_j)$ are the auxiliary functions that depend on parameters $C_j$. Now, the final solution is expressed in the following form:

$$\Phi(x,t,C_j;q) = u_0(x,t) + \sum_{i=1}^{\infty} u_i(x,t,C_j)q^i \tag{19}$$

Substituting Equation (19) into Equation (14), and equating the like powers of $q$, the following equations are obtained [where $q^0$ corresponds to Equations (16) and (17)]:

$$\mathcal{L}(u_1(x,t,C_j)) = H_1(x,t,C_j)\mathcal{N}_0(u_0(x,t)) \tag{20}$$

subject to the boundary condition:

$$\mathcal{B}\left(u_1, \frac{\partial u_1}{\partial x}, \frac{\partial u_1}{\partial t}\right) = 0 \tag{21}$$

For $k = 2,3,4,\ldots,$

$$\begin{aligned}
\mathcal{L}[u_k(x,t,C_j) \quad &-u_{k-1}(x,t,C_j)] \\
&= H_k(x,t,C_j)\mathcal{N}_0(u_0(x,t)) \\
&+ \sum_{j=1}^{k-1} H_j(x,t,C_j)\left[\mathcal{L}\left[u_{k-j}(x,t,C_j)\right]\right. \\
&+ \left. \mathcal{N}_{k-j}\left[u_0(x,t), u_1(x,t,C_j),\ldots, u_{k-j}(x,t,C_j)\right]\right]
\end{aligned} \tag{22}$$

subject to the boundary conditions:

$$\mathcal{B}\left(u_k, \frac{\partial u_k}{\partial x}, \frac{\partial u_k}{\partial t}\right) = 0 \tag{23}$$

where the term $\mathcal{N}_{k-j}\left[u_0(x,t), u_1(x,t,C_j),\ldots, u_{k-j}(x,t,C_j)\right]$ is the coefficient of $q^m$, which is obtained by expanding $\mathcal{N}(\Phi(x,t,C_j;q))$ as follows:

$$\begin{aligned}
\mathcal{N}(\Phi(x,t,C_j;q)) \\
&= \mathcal{N}_0(u_0(x,t)) + q\mathcal{N}_1(u_0(x,t), u_1(x,t,C_j)) \\
&+ q^2 \mathcal{N}_2(u_0(x,t), u_1(x,t,C_j), u_2(x,t,C_j)) + \ldots
\end{aligned} \tag{24}$$

It may be noted that the choice of auxiliary functions $H_j(x,t,C_j)$ strongly influences the convergence of the series Equation (19). According to [20], $H_j(x,t,C_j)$ should be chosen in such a way that the product $H_j(x,t,C_j)[\mathcal{L}[u_{k-j}(x,t,C_j)] + \mathcal{N}_{k-j}[u_0(x,t), u_1(x,t,C_j),\ldots,$ $u_{k-j}(x,t,C_j)]]$ and $H_j(x,t,C_j)$ are of the same form. Now, if the series Equation (19) converges at $q = 1$, then we have:

$$u(x,t,C_j) = u_0(x,t) + \sum_{j=1}^{\infty} u_j(x,t,C_j) \tag{25}$$

Finally, the approximate solution can be obtained by considering a finite number of terms of the series Equation (25). The choices for parameters $C_j$ and auxiliary function will be discussed in the next section.

### 3. Governing Equation and Solution Methodologies

Let us consider a horizontal confined aquifer having a constant thickness, which is infinitely extended horizontally and is homogeneous and isotropic. Some of the further assumptions are that the aquifer has a single pumping well with a constant rate with respect to time and a negligibly small diameter. Further, it is assumed that the well penetrates the entire aquifer, and the hydraulic head in the aquifer, before pumping, is uniform throughout the aquifer. The continuity equation and Darcy's law are combined together to obtain the following governing partial differential equation, representing saturated flow in a horizontal confined aquifer [21]:

$$\frac{\partial^2 s}{\partial x^2} + \frac{\partial^2 s}{\partial y^2} = \frac{S}{T}\frac{\partial s}{\partial t} \tag{26}$$

where $x$ and $y$ are the spatial variables; $t$ is the temporal variable; $s$ is the hydraulic head; and $T$ and $S$ are the transmittivity and storativity, respectively. It is convenient to convert Equation (26) into radial coordinates because of the radial symmetry of the hydraulic-head drawdown around a well. Therefore, using $r = \sqrt{x^2 + y^2}$, where $r$ is the radial coordinate, the governing equation becomes:

$$\frac{\partial^2 s}{\partial r^2} + \frac{1}{r}\frac{\partial s}{\partial r} = \frac{S}{T}\frac{\partial s}{\partial t} \tag{27}$$

The radial coordinate allows us to convert two space dimensions into a single coordinate, specifically a one-dimensional line starting from the well center at $r = 0$ to the infinite extremity at $r = \infty$. The flow configuration is schematically provided in Figure 1.

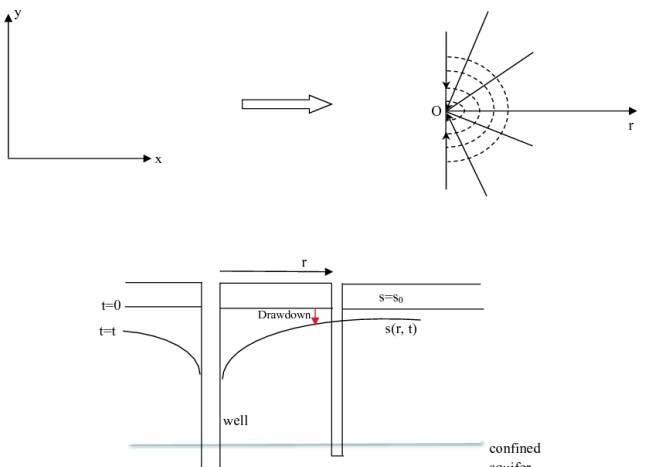

**Figure 1.** Radial flow to a well in a horizontal confined aquifer.

The initial condition is given as:

$$s(r,0) = s_0 \text{ for all } r \tag{28}$$

where $s_0$ is the constant initial hydraulic head. For the boundary conditions, no hydraulic-head drawdown is assumed at the infinite extremity, i.e.,

$$s(\infty, t) = s_0 \text{ for all } t \tag{29}$$

Again, a constant pumping rate is assumed at the well:

$$\lim_{r \to 0}\left( r\frac{\partial s}{\partial r} \right) = \frac{Q}{2\pi T} \text{ for all } t \tag{30}$$

The second boundary condition Equation (30) is invoked by the application of Darcy's law at the well face. The solution of Equation (27), together with the initial and boundary conditions Equations (28)–(30), provides the hydraulic head at any radial distance $r$ and time $t$ after the start of pumping. In practice, the solutions are often presented in terms of the drawdown expressed in the head $s_0 - s(r, t)$.

### 3.1. Theis Solution

An analytical solution for the governing Equation (27), together with the given conditions Equations (28)–(30), was first provided by Theis [1], who followed an analogy with heat conduction in solids to arrive at the desired solution. The derived solution can be expressed as:

$$s_0 - s(r, t) = \frac{Q}{4\pi T} \int_v^\infty \frac{\exp(-v)}{v} dv \tag{31}$$

where $v = \frac{r^2 S}{4Tt}$. The solution Equation (31) is popularly known as the *Theis solution*. It may be noted that the governing PDE can be solved using different techniques, such as the Laplace transform, Fourier transform, similarity transformation, etc. In the mathematics literature, the integral on the right side of Equation (31) is known as the *exponential integral*. However, in relation to the problem considered here, it is generally called the *well function* and is denoted by $W(v)$. Accordingly, Equation (31) becomes:

$$s_0 - s(r, t) = \frac{Q}{4\pi T} W(v) \tag{32}$$

The function $W(u)$ follows several interesting properties, e.g.,

$$W(-\infty) = -\infty, \; W(0) = +\infty, \; W(+\infty) = 0, \; W(v) = \Gamma(0, v) \tag{33}$$

where $\Gamma(x, v)$ is the upper incomplete gamma function, defined as:

$$\Gamma(x, v) = \int_v^\infty t^{x-1} \exp(-t) dt \tag{34}$$

There are two convergent series for the integral $W(v)$. One of them can be expressed as follows ([4,8]):

$$W(v) = -\gamma - \ln v + \sum_{i=1}^\infty \frac{(-1)^{i+1} v^i}{i \, i!} \qquad |Arg(v)| < \pi \tag{35}$$

where $\gamma$ is the Euler–Mascheroni constant, and its value (up to four decimal places) is 0.5772. For calculating the well function using Equation (35), there are some drawbacks. For example, this series produces inaccurate results for $v > 2.5$ if one considers a few terms. This occurs due to the cancellation. Further, the convergence rate of the series is slow, hence, the approximation may not be desirable in practical situations. A more rapidly convergent series, attributed to S. Ramanujan, is given by [22]:

$$Ei(v) = \gamma + \ln v + \exp(v/2) \sum_{k=1}^\infty \frac{(-1)^{k-1} v^k}{k! 2^{k-1}} \sum_{n=0}^{\lfloor (k-1)/2 \rfloor} \frac{1}{2n+1} \tag{36}$$

where $\lfloor (k-1)/2 \rfloor$ denotes the *floor function*. The integral $W(v)$ is related to Equation (36) as follows:

$$W(v) = -Ei(-v) \tag{37}$$

Equation (37) provides an accurate estimate for the integral for small values of $v$. However, both Equations (35) and (36) are not useful for larger values of $v$. For large values of $v$, there is an asymptotic series approximation as follows [5]:

$$W(v) = \frac{\exp(-v)}{v}\left(\sum_{k=0}^{n-1}\frac{k!}{(-v)^k} + \mathcal{O}\left(|v|^{-n}\right)\right)$$

(38)

where $\mathcal{O}$ denotes the 'Big-O' notation. Equation (38) can be obtained by expanding the integral of $W(v)$ by parts.

The previous discussions are based on the series approximation for the integral and found to be accurate for either small or large values of $v$. However, for practical applications, one needs to have an accurate expression for $W(v)$ for a wide range of $v$. To that end, several approximations have been proposed in the literature. Here, we mention some of them. Swamee and Ojha [12] proposed the following approximation:

$$W(v) = \left[f_1(v)^{-7.7} + f_2(v)\right]^{-0.13}$$

(39)

where

$$f_1(v) = \ln\left[\left(0.65 + \frac{0.56146}{v}\right)(1+v)\right]$$

(40)

$$f_2(v) = v^4\exp(7.7v)(2+v)^{3.7}$$

(41)

Barry et al. [13] offered the following full-range solution:

$$W(v) = \frac{\exp(-v)}{a_1 + (1-a_1)\exp\left(-\frac{v}{1-a_1}\right)}\ln\left[1 + \frac{a_1}{v} - \frac{1-a_1}{(f_3(v) + a_2 v)^2}\right]$$

(42)

where

$$a_1 = \exp(-\gamma),\ a_2 = \sqrt{\frac{2(1-a_1)}{a_1(2-a_1)}},\ \gamma\text{ is the Euler–Mascheroni constant}$$

(43)

$$f_3(v) = \frac{1}{1+v\sqrt{v}} + \frac{\hat{a}_2\widetilde{a_2}}{1+\widetilde{a_2}},\ \hat{a}_2 = \frac{(1-a_1)\left(a_1^2 - 6a_1 + 12\right)}{3a_1(2-a_1)^2 a_2},\ \widetilde{a_2} = \frac{20}{47}v^{\sqrt{\frac{31}{26}}}$$

(44)

In a recent study, Vatankhah [14] proposed the following approximation:

$$W(v) = \left\{\left[\left(1 + b_1 v^{b_2}\right)\ln\left(\frac{b_3}{v} + b_4\right)\right]^{-p} + \left[\frac{1}{v\exp(v)}\left(\frac{v+b_5}{v+b_6}\right)\right]^{-p}\right\}^{-\frac{1}{p}}$$

(45)

where $p = 2$, $b_1 = -0.19$, $b_2 = 0.7$, $b_3 = 0.565$, $b_4 = 4$, $b_5 = 0.444$, and $b_6 = 1.384$.

### 3.2. Homotopy-Based Solutions

It may be noted that the homotopy-based methods are easier to apply to ODEs than the PDEs. This is the case because the PDE involves more than one independent variable, which makes it difficult to choose the operators and initial approximations for these methods. Therefore, first, we convert the governing PDE Equation (27) into an ODE, using the following similarity transformation:

$$v = \frac{r^2 S}{4tT}$$

(46)

Then

$$\frac{\partial s}{\partial r} = \frac{ds}{dv}\frac{\partial v}{\partial r} = \frac{2rS}{4tT}\frac{ds}{dv} \Rightarrow \frac{1}{r}\frac{\partial s}{\partial r} = \frac{2S}{4tT}\frac{ds}{dv}$$

(47)

$$\frac{\partial^2 s}{\partial r^2} = \frac{\partial}{\partial r}\left[\frac{2rS}{4tT}\frac{ds}{dv}\right] = \frac{2S}{4tT}\frac{ds}{dv} + \frac{4r^2S^2}{16t^2T^2}\frac{d^2s}{dv^2} \tag{48}$$

$$\frac{\partial s}{\partial t} = \frac{ds}{dv}\frac{\partial v}{\partial t} = -\frac{r^2S}{4t^2T}\frac{ds}{dv} \tag{49}$$

Using Equations (29)–(47), the governing Equation (27) becomes:

$$v\frac{d^2s}{dv^2} + (1+v)\frac{ds}{dv} = 0 \tag{50}$$

Following Equation (46), the conditions Equations (28) and (29) become:

$$s(v \to \infty) = s_0 \tag{51}$$

and Equation (30) changes to:

$$\lim_{r\to 0}\left(r\frac{\partial s}{\partial r}\right) = \frac{Q}{2\pi T} \Rightarrow \lim_{v\to 0}\left(v\frac{ds}{dv}\right) = \frac{Q}{4\pi T} \tag{52}$$

To be in line with Theis's method, we intend to obtain the solution as $s_0 - s(v)$. For that, we use the transformation:

$$s(v) = s_0 + \bar{s}(v) \tag{53}$$

Accordingly, Equation (50) becomes:

$$v\frac{d^2\bar{s}}{dv^2} + (1+v)\frac{d\bar{s}}{dv} = 0 \tag{54}$$

The boundary conditions (51) and (52) become:

$$\bar{s}(v \to \infty) = 0 \tag{55}$$

$$\lim_{v\to 0}\left(v\frac{d\bar{s}}{dv}\right) = -\frac{Q}{4\pi T} \tag{56}$$

3.2.1. HAM-Based Solution

Here, we apply HAM to Equation (54) together with the conditions Equations (55) and (56). Following the discussion in Section 2.1, we consider the nonlinear operator for the problem as follows:

$$\mathcal{N}[\Phi(v;q)] = v\frac{\partial^2\Phi(v;q)}{\partial v^2} + (1+v)\frac{\partial\Phi(v;q)}{\partial v} \tag{57}$$

It may be noted that Equation (57) is the original governing equation. Using Equation (57), terms $R_m$ are obtained as follows:

$$R_m\left(\overset{\to}{\bar{s}}_{m-1}\right) = v\frac{d^2\bar{s}_{m-1}}{dv^2} + (1+v)\frac{d\bar{s}_{m-1}}{dv} \tag{58}$$

Now, we define the base functions to represent the solution of the governing equation. To that end, the following set of base functions is chosen for the considered problem:

$$\{\exp(-v)\ln(v) + [W(v) + \exp(-v)]v^m \mid m = 0, 1, 2, \ldots\} \tag{59}$$

so that

$$\bar{s}(v) = a_0 \exp(-v)\ln(v) + [W(v) + \exp(-v)]\sum_{n=0}^{\infty} b_n(v)^n \tag{60}$$

where $a_0$ and $b_m$ are the coefficients of the series. Equation (60) provides the so-called *rule of solution expression.* Following the rule of solution expression, the linear operator and the initial approximation are chosen, respectively, as follows:

$$\mathcal{L}[\Phi(v;q)] = \frac{\partial^2 \Phi(v;q)}{\partial v^2} \quad \text{with the property} \quad \mathcal{L}[C_2 v + C_3] = 0 \tag{61}$$

$$\bar{s}_0(v) = -\frac{Q}{4\pi T} \exp(-v) \ln(v) \tag{62}$$

where $C_2$ and $C_3$ are integral constants. It may be noted that the initial approximation Equation (62) is chosen in accordance with the given boundary conditions Equations (55) and (56). Using Equation (61), the higher-order terms can be obtained from Equation (62) as follows:

$$\bar{s}_m(v) = \chi_m \bar{s}_{m-1}(v) + \hbar \int_0^v \int_0^v H(x) R_m \left( \vec{\bar{s}}_{m-1} \right) dx \, dy + C_2 v + C_3, \quad m = 1, 2, 3, \ldots \tag{63}$$

where $R_m$ is given by Equation (68), and constants $C_2$ and $C_3$ can be determined from the boundary conditions for the higher-order deformation equations given by Equation (63).

Now, the *rule of coefficient ergodicity* determines the auxiliary function $H(v)$. It may be noted that the function $H(v)$ can be of many forms, based on the rule of solution expression Equation (60). However, for simplicity, one can select $H(v) = 1$ in order to avoid unnecessary difficulty in computation. Further, this is consistent with the theory given in [23]. Finally, the approximate solution can be obtained as follows:

$$\bar{s}(v) = s_0 - s(v) \approx \bar{s}_0(v) + \sum_{n=1}^M \bar{s}_n(v) \tag{64}$$

### 3.2.2. HPM-Based Solution

Following the discussion in Section 2.2, we select the linear and nonlinear operators as follows:

$$\mathcal{L}(\Phi(v;q)) = \frac{\partial^2 \Phi(v;q)}{\partial v^2} \tag{65}$$

$$\mathcal{N}(\Phi(v;q)) = v \frac{\partial^2 \Phi(v;q)}{\partial v^2} + (1+v) \frac{\partial \Phi(v;q)}{\partial v} \tag{66}$$

Using these operators, Equation (9) takes on the form:

$$(1-q)\left[ \frac{\partial^2 \Phi(v;q)}{\partial v^2} - \frac{d^2 \bar{s}_0}{dv^2} \right] + q\left[ v \frac{\partial^2 \Phi(v;q)}{\partial v^2} + (1+v) \frac{\partial \Phi(v;q)}{\partial v} \right] = 0 \tag{67}$$

Substituting the series expression Equation (10) into Equation (67), we have:

$$(1-q)\left[ \frac{\partial^2 \Phi(v;q)}{\partial v^2} - \frac{d^2 \bar{s}_0}{dv^2} \right]$$
$$= \left( \frac{d^2 \Phi_0}{dv^2} - \frac{d^2 \bar{s}_0}{dv^2} \right) + q\left( \frac{d^2 \Phi_1}{dv^2} - \left( \frac{d^2 \Phi_0}{dv^2} - \frac{d^2 \bar{s}_0}{dv^2} \right) \right) + q^2 \left( \frac{d^2 \Phi_2}{dv^2} - \frac{d^2 \Phi_1}{dv^2} \right) + q^3 \left( \frac{d^2 \Phi_3}{dv^2} - \frac{d^2 \Phi_2}{dv^2} \right) \tag{68}$$
$$+ q^4 \left( \frac{d^2 \Phi_4}{dv^2} - \frac{d^2 \Phi_3}{dv^2} \right) + \ldots$$

$$v \frac{\partial^2 \Phi(v;q)}{\partial v^2} + (1+v) \frac{\partial \Phi(v;q)}{\partial v}$$
$$= \left[ v \frac{d^2 \Phi_0}{dv^2} + (1+v) \frac{d\Phi_0}{dv} \right] + q\left[ v \frac{d^2 \Phi_1}{dv^2} + (1+v) \frac{d\Phi_1}{dv} \right] \tag{69}$$
$$+ q^2 \left[ v \frac{d^2 \Phi_2}{dv^2} + (1+v) \frac{d\Phi_2}{dv} \right] + q^3 \left[ v \frac{d^2 \Phi_3}{dv^2} + (1+v) \frac{d\Phi_3}{dv} \right] + \ldots$$

Using the boundary conditions $\bar{s}(v \to \infty) = 0$ and $\lim_{v \to 0}\left(v\frac{d\bar{s}}{dv}\right) = -\frac{Q}{4\pi T}$, we have:

$$\Phi_0(v \to \infty) = 0, \ \Phi_1(v \to \infty) = 0, \ \Phi_2(v \to \infty) = 0, \ \ldots. \tag{70}$$

$$\lim_{v \to 0}\left(v\frac{d\Phi_0}{dv}\right) = -\frac{Q}{4\pi T}, \ \lim_{v \to 0}\left(v\frac{d\Phi_1}{dv}\right) = 0, \ \lim_{v \to 0}\left(v\frac{d\Phi_2}{dv}\right) = 0, \ \ldots \tag{71}$$

Using Equations (70) and (71) and equating the like powers of $q$ in Equation (67), the following system of differential equations is obtained:

$$\frac{d^2\Phi_0}{dv^2} - \frac{d^2\bar{s}_0}{dv^2} = 0 \text{ subject to } \Phi_0(v \to \infty) = 0, \ \lim_{v \to 0}\left(v\frac{d\Phi_0}{dv}\right) = -\frac{Q}{4\pi T} \tag{72}$$

$$\frac{d^2\Phi_1}{dv^2} - \left(\frac{d^2\Phi_0}{dv^2} - \frac{d^2\bar{s}_0}{dv^2}\right) + v\frac{d^2\Phi_0}{dv^2} + (1+v)\frac{d\Phi_0}{dv} = 0 \text{ subject to } \Phi_1(v \to \infty) = 0, \ \lim_{v \to 0}\left(v\frac{d\Phi_1}{dv}\right) = 0 \tag{73}$$

$$\frac{d^2\Phi_2}{dv^2} - \frac{d^2\Phi_1}{dv^2} + v\frac{d^2\Phi_1}{dv^2} + (1+v)\frac{d\Phi_1}{dv} = 0 \text{ subject to } \Phi_2(v \to \infty) = 0, \ \lim_{v \to 0}\left(v\frac{d\Phi_2}{dv}\right) = 0 \tag{74}$$

$$\frac{d^2\Phi_3}{dv^2} - \frac{d^2\Phi_2}{dv^2} + v\frac{d^2\Phi_2}{dv^2} + (1+v)\frac{d\Phi_2}{dv} = 0 \text{ subject to } \Phi_3(v \to \infty) = 0, \ \lim_{v \to 0}\left(v\frac{d\Phi_3}{dv}\right) = 0 \tag{75}$$

$$\frac{d^2\Phi_4}{dv^2} - \frac{d^2\Phi_3}{dv^2} + v\frac{d^2\Phi_3}{dv^2} + (1+v)\frac{d\Phi_3}{dv} = 0 \text{ subject to } \Phi_4(v \to \infty) = 0, \ \lim_{v \to 0}\left(v\frac{d\Phi_4}{dv}\right) = 0 \tag{76}$$

Proceeding in a like manner, one can arrive at the following recurrence relation:

$$\frac{d^2\Phi_m}{dv^2} = (1-v)\frac{d^2\Phi_{m-1}}{dv^2} - (1+v)\frac{d\Phi_{m-1}}{dv} \text{ subject to } \Phi_m(v \to \infty) = 0, \ \lim_{v \to 0}\left(v\frac{d\Phi_m}{dv}\right) = 0 \text{ for } m \geq 2 \tag{77}$$

The initial approximation can be chosen as $\Phi_0 = -\frac{Q}{4\pi T}\exp(-v)\ln v$. Using this initial approximation, we can solve the equations iteratively using symbolic software. Finally, the HPM-based solution can be approximated as follows:

$$\bar{s}(v) \approx \sum_{i=0}^{M} \Phi_i \tag{78}$$

3.2.3. OHAM-Based Solution

When dealing with the problem using OHAM, it is observed that due to boundary conditions, it may not be possible to solve the governing Equation (54) using OHAM. To that end, first, we convert the boundary value problem Equation (54) into an ODE, as follows:

$$v\frac{d^2\bar{s}}{dv^2} + (1+v)\frac{d\bar{s}}{dv} = 0 \Rightarrow \frac{d}{dv}\left(v\frac{d\bar{s}}{dv}\right) + v\frac{d\bar{s}}{dv} = 0 \Rightarrow \frac{du}{dv} + u = 0 \tag{79}$$

subject to

$$\lim_{v \to 0}(u) = -\frac{Q}{4\pi T} \tag{80}$$

Solving Equation (79) together with Equation (80), one obtains:

$$v\frac{d\bar{s}}{dv} + \frac{Q}{4\pi T}\exp(-v) = 0 \quad \text{subject to} \quad \bar{s}(\infty) = 0 \tag{81}$$

For applying OHAM, Equation (81) can be written in the following form:

$$\mathcal{L}[\bar{s}(v)] + \mathcal{N}[\bar{s}(v)] + h(v) = 0 \tag{82}$$

We select $\mathcal{L}[\bar{s}(v)] = \frac{d\bar{s}}{dv}$, $\mathcal{N}[\bar{s}(v)] = (v-1)\frac{d\bar{s}}{dv}$, and $h(v) = \frac{Q}{4\pi T}\exp(-v)$. Using these expressions, solution of the zeroth-order representation Equation (16) becomes:

$$\bar{s}_0(v) = \frac{Q}{4\pi T}\exp(-v) \tag{83}$$

The following relation is obtained for the expressions of $\mathcal{N}_0, \mathcal{N}_1, \mathcal{N}_2$, etc.:

$$(v-1)\frac{d\bar{s}}{dv} = \left[(v-1)\frac{d\bar{s}_0}{dv}\right] + q\left[(v-1)\frac{d\bar{s}_1}{dv}\right] + q^2\left[(v-1)\frac{d\bar{s}_2}{dv}\right] + q^3\left[(v-1)\frac{d\bar{s}_3}{dv}\right] + \dots \tag{84}$$

Using Equation (84), the first-order representation Equation (20) reduces to:

$$\frac{d\bar{s}_1}{dv} = H_1(v, C_i)\left\{(v-1)\frac{d\bar{s}_0}{dv}\right\} \text{ subject to } \bar{s}_1(v \to \infty) = 0 \tag{85}$$

The auxiliary function is chosen as $H_1(v, C_i) = C_1 + C_2\exp(-v)$. Putting $k = 2$ and $H_2(v, C_i) = C_3 + C_4\exp(-v)$, the second-order representation Equation (22) becomes:

$$\frac{d\bar{s}_2}{dv} = \frac{d\bar{s}_1}{dv} + (C_3 + C_4\exp(-v))\left[(v-1)\frac{d\bar{s}_0}{dv}\right] + (C_1 + C_2\exp(-v))\left[\frac{d\bar{s}_1}{dv} + (v-1)\frac{d\bar{s}_1}{dv}\right] \text{ subject to } \bar{s}_2(v \to \infty) = 0 \tag{86}$$

The first three terms of the OHAM-based solution can be obtained by solving the above equations using symbolic computation software, such as MATLAB. Further, we restrict our analysis up to $k = 2$, as the objective is to obtain an accurate solution with just a few terms of the OHAM-based series. Finally, the approximate solution can be found as:

$$\bar{s}(v) \approx \bar{s}_0(v) + \bar{s}_1(v, C_1, C_2) + \bar{s}_2(v, C_1, C_2, C_3, C_4) \tag{87}$$

where the terms are given by Equations (83), (85), and (86).

## 4. Results and Discussion

Here, first, we discuss the validity of the approximations for the well function. Then, the HAM-, HPM-, and OHAM-based approximations are verified with the numerical solution to the problem. Finally, the proposed solutions are compared with the existing approximations in order to have a numerical assessment.

### 4.1. Validation of the Well Function's Approximations

In Figure 2, we validate the series approximation given by Equation (35) by comparing it with the numerical solution of the corresponding exponential integral. The numerical values of $W(v)$ were calculated using the MATLAB script 'integral', which uses the global adaptive quadrature rule [24]. We compared 10, 20, and 40 terms of the series Equation (35) and observe that one needs more than 40 terms of the series to obtain an accurate approximation for $v$ up to 6.5. Further, the convergence rate is too slow as we increase the order of approximation.

Figure 3 compares the numerical solution and the series approximation (38) of $W(v)$, considering some terms of the series. The series is an asymptotic expansion that is valid for large values of $v$. As expected, it is observed from the figure that the series performs well only for large values of $v$. However, it produces inaccurate results for small values of $v$. For a comparative assessment, we plot together the numerical solution and both the series Equations (35) and (38) in Figure 4. It is observed from the figure that one of them performs well for smaller values, and the other is accurate only for large values of $v$.

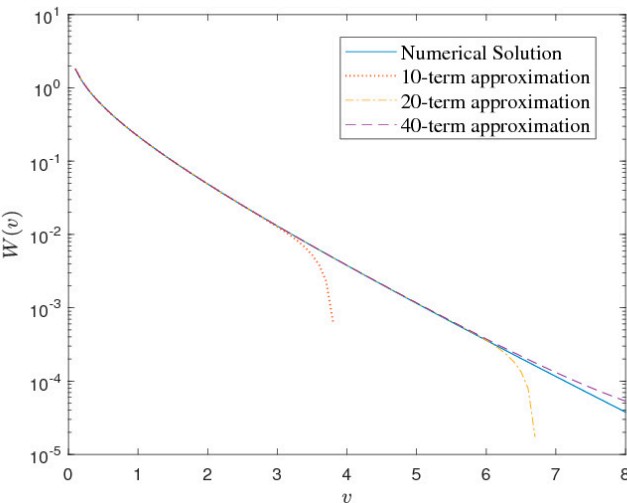

**Figure 2.** Comparison between numerical solution and 10, 20, and 40 terms of the series Equation (35) for $W(v)$.

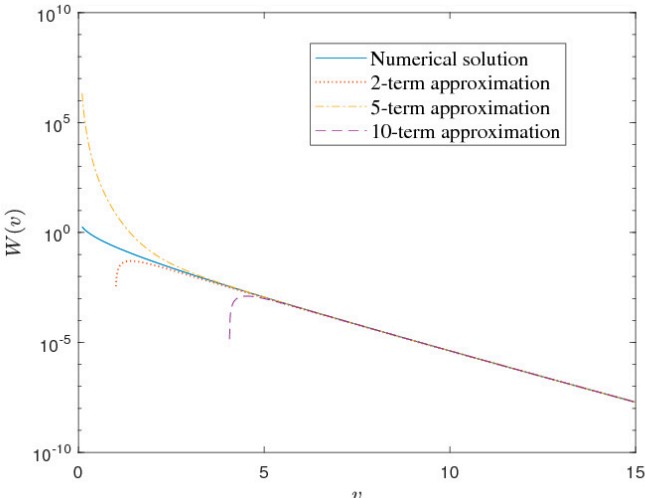

**Figure 3.** Comparison between numerical solution and 2, 5, and 100 terms of the series Equation (38) for $W(v)$.

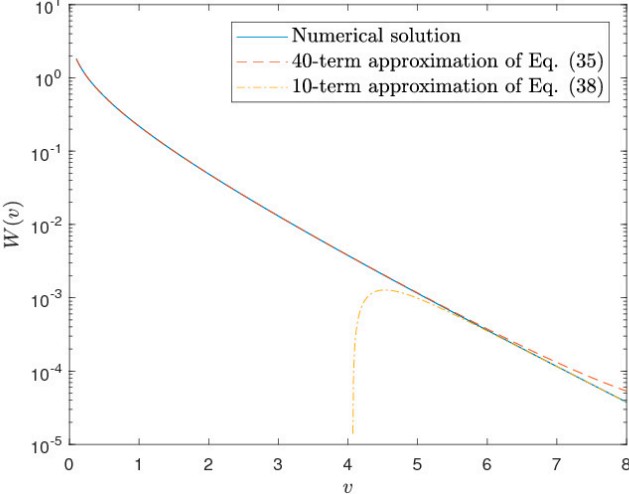

**Figure 4.** Comparison between numerical solution, 40 terms of the series Equation (35), and 10 terms of the series Equation (38) for $W(v)$.

### 4.2. Numerical Convergence and Validation of the HAM-Based Solution

In the framework of HAM, the auxiliary parameter ℏ plays the key role in determining the convergence of the series solution. An optimal value for the parameter can be obtained by minimizing the following squared residual error of Equation (54), which can be calculated as follows:

$$\Delta_m = \int\limits_{v \in \Omega} (\mathcal{N}[\bar{s}(v)])^2 dv \tag{88}$$

where $\Omega$ is the domain of the equation. The HAM-based series solution converges when the corresponding residual error Equation (88) becomes zero. A test case is considered here, where the parameters were chosen as $Q = 4 \times 10^{-3}$ m$^3$s$^{-1}$, $T = 0.0023$ m$^2$s$^{-1}$, and $S = 7.5 \times 10^{-4}$. Using these parameter values, we assessed the HAM-based solution by calculating the squared residual errors for different orders of approximations. The squared residual errors for different orders of approximation are plotted in Figure 5, where it is seen that the error decreases with the increasing order of approximation. Thus, the numerical convergence was established, and the choice of operators and parameters was validated. For a quantitative assessment, the numerical results are also reported in Table 1, along with the computational time taken by the computer to produce the corresponding order of approximations. It can be seen from the table that even though HAM is an analytical series approximation technique, it still does not involve time complexity. On the other hand, for the selected case, we compared the Theis solution (with the integral computed using the MATLAB script 'integral', which uses the global adaptive quadrature rule [24]) and the 10th-order HAM-based approximate solution in Figure 6. It may be noted that the same method was used for the numerical solution related to HPM and OHAM. An excellent agreement is found between the computed and observed values. Additionally, for a comparative idea, 4th-, 7th-, and 10th-order approximations were considered and compared with the Theis solution, as shown in Table 2. It can be observed from the table that the higher the order of approximation, the better the accuracy. All computations were performed using the BVPh 2.0 package developed by [25]. A flowchart containing the steps of HAM for the present problem is provided in Figure 7. The theoretical convergence analysis is provided in Appendix A.

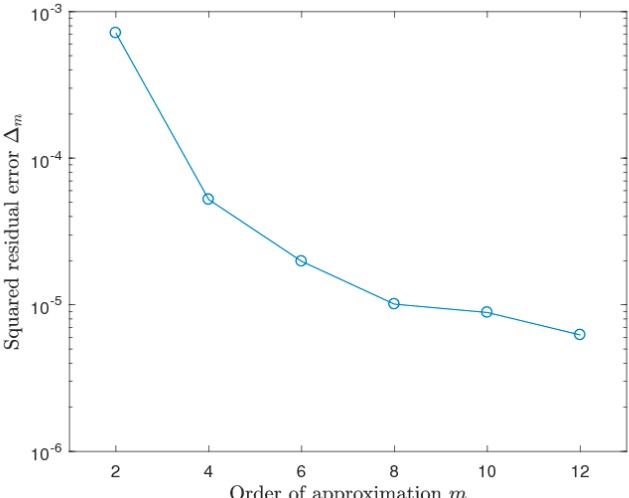

**Figure 5.** Squared residual error ($\Delta_m$) versus different orders of approximations ($m$) of the HAM-based solution for the selected case.

**Table 1.** Squared residual error ($\Delta_m$) and computational time versus different orders of approximation (m) for the selected case.

| Order of Approximation (m) | Squared Residual Error ($\Delta_m$) | Computational Time (s) |
| --- | --- | --- |
| 2 | $7.15 \times 10^{-4}$ | 0.212 |
| 4 | $5.23 \times 10^{-5}$ | 1.135 |
| 6 | $1.98 \times 10^{-5}$ | 2.481 |
| 8 | $1.01 \times 10^{-5}$ | 5.074 |
| 10 | $8.86 \times 10^{-6}$ | 6.843 |
| 12 | $6.23 \times 10^{-6}$ | 10.149 |

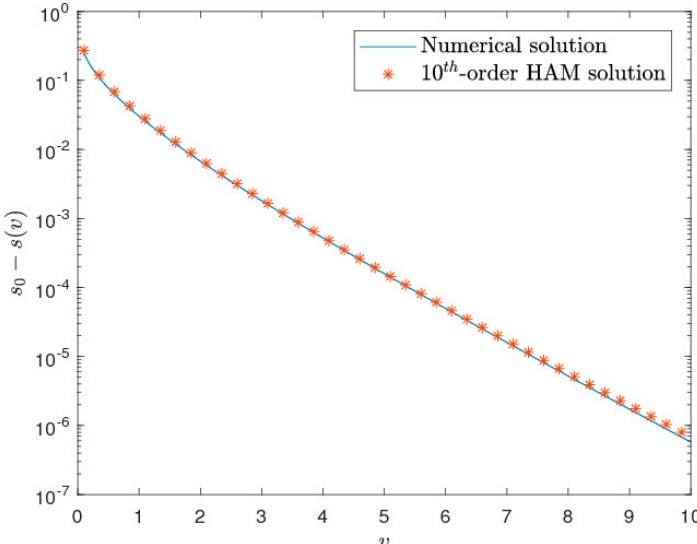

**Figure 6.** Comparison between the Theis solution and 10th-order HAM-based solution for the selected case.

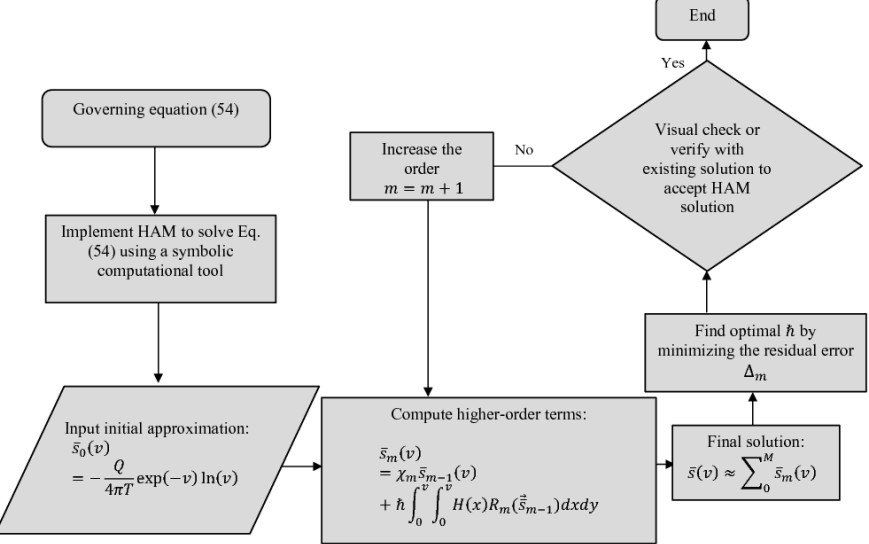

**Figure 7.** Flowchart for the HAM solution (64).

**Table 2.** Comparison between HAM-based approximation and numerical solution for the selected case.

| $u$ | Numerical Solution | HAM-Based Approximation | | |
| --- | --- | --- | --- | --- |
| | | 4th Order | 7th Order | 10th Order |
| 0.1 | $2.523 \times 10^{-1}$ | $3.009 \times 10^{-1}$ | $2.821 \times 10^{-1}$ | $2.726 \times 10^{-1}$ |
| 1 | $3.036 \times 10^{-2}$ | $3.995 \times 10^{-2}$ | $3.548 \times 10^{-2}$ | $3.305 \times 10^{-2}$ |
| 2 | $6.768 \times 10^{-3}$ | $9.302 \times 10^{-3}$ | $8.011 \times 10^{-3}$ | $7.228 \times 10^{-3}$ |
| 3 | $1.806 \times 10^{-3}$ | $2.560 \times 10^{-3}$ | $2.146 \times 10^{-3}$ | $1.886 \times 10^{-3}$ |
| 4 | $5.230 \times 10^{-4}$ | $7.603 \times 10^{-4}$ | $6.213 \times 10^{-4}$ | $5.377 \times 10^{-4}$ |
| 5 | $1.589 \times 10^{-4}$ | $2.359 \times 10^{-4}$ | $1.881 \times 10^{-4}$ | $1.627 \times 10^{-4}$ |
| 6 | $4.983 \times 10^{-5}$ | $7.530 \times 10^{-5}$ | $5.870 \times 10^{-5}$ | $5.150 \times 10^{-5}$ |
| 7 | $1.598 \times 10^{-5}$ | $2.453 \times 10^{-5}$ | $1.871 \times 10^{-5}$ | $1.689 \times 10^{-5}$ |
| 8 | $5.213 \times 10^{-6}$ | $8.111 \times 10^{-6}$ | $6.066 \times 10^{-6}$ | $5.688 \times 10^{-6}$ |
| 9 | $1.723 \times 10^{-6}$ | $2.713 \times 10^{-6}$ | $1.993 \times 10^{-6}$ | $1.955 \times 10^{-6}$ |
| 10 | $5.753 \times 10^{-7}$ | $9.160 \times 10^{-7}$ | $6.617 \times 10^{-7}$ | $6.814 \times 10^{-7}$ |

*4.3. Validation of HPM-Based Solution*

For the selected case, the HPM-based analytical solution was validated over the solution given by [1], where the integral is performed numerically. It may be noted that, unlike HAM, HPM solutions are often valid only within a small domain [16]. Therefore, for our case here, we considered the domain $v \in [0.1, 2]$. After assessing the solution within this domain, we observe that the series with four terms is more accurate than the other lower-order terms. The comparison between the Theis solution and the four terms of HPM approximation is given in Figure 8, where it can be seen that as the domain increases, the accuracy of the solution decreases. This is a default problem with HPM, as the methodology does not contain any convergence-control parameter like HAM. Indeed, one may try with different combinations of initial approximations and operators, which might then work in producing a more accurate solution to the problem. Table 3 shows a numerical comparison between the HPM-based values and the Theis solution. For the convenience of the readers, a flowchart containing the steps of HPM is provided in Figure 9.

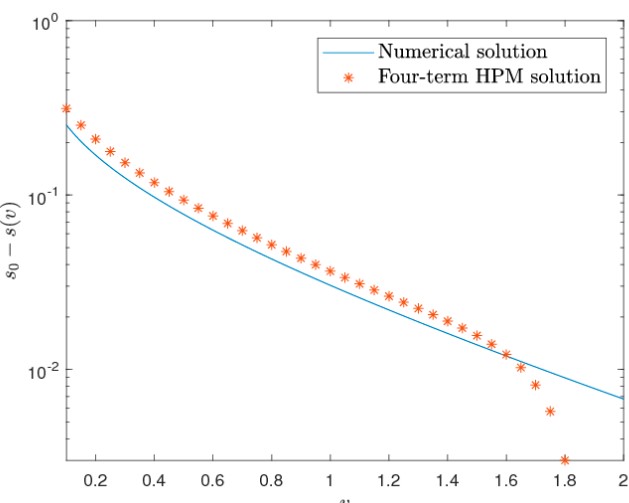

**Figure 8.** Comparison between the Theis solution and the four terms of HPM-based solution for the selected case.

**Table 3.** Comparison between four terms of the HPM-based approximation and Theis solution for the selected case.

| $u$ | Numerical Solution | Four Terms of the HPM-Based Approximation |
|---|---|---|
| 0.1 | $2.523 \times 10^{-1}$ | $3.135 \times 10^{-1}$ |
| 0.3 | $1.253 \times 10^{-1}$ | $1.533 \times 10^{-1}$ |
| 0.5 | $7.747 \times 10^{-2}$ | $9.362 \times 10^{-2}$ |
| 0.7 | $5.173 \times 10^{-2}$ | $6.246 \times 10^{-2}$ |
| 0.9 | $3.601 \times 10^{-2}$ | $4.346 \times 10^{-2}$ |
| 1.1 | $2.574 \times 10^{-2}$ | $3.097 \times 10^{-2}$ |
| 1.3 | $1.875 \times 10^{-2}$ | $2.237 \times 10^{-2}$ |
| 1.5 | $1.384 \times 10^{-2}$ | $1.563 \times 10^{-2}$ |
| 1.7 | $1.033 \times 10^{-2}$ | $8.139 \times 10^{-3}$ |
| 2.0 | $6.768 \times 10^{-3}$ | $1.298 \times 10^{-2}$ |

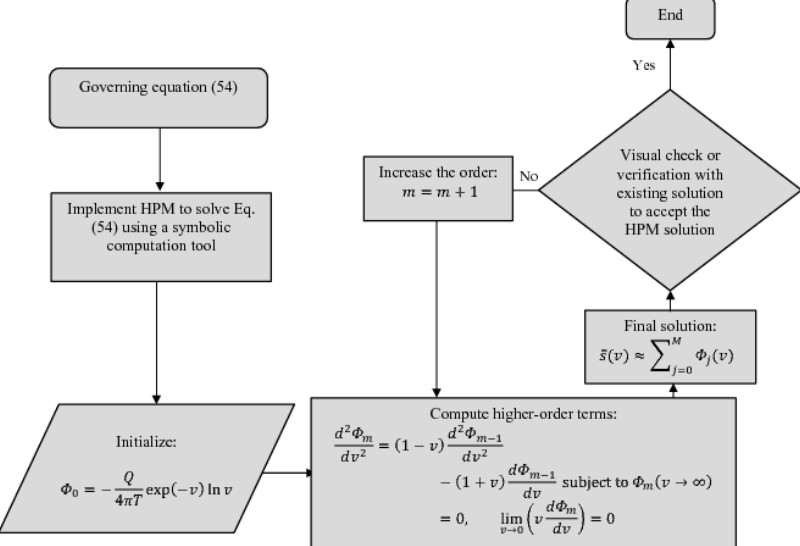

**Figure 9.** Flowchart for the HPM solution (78).

*4.4. Validation of OHAM-Based Solution*

It can be seen from Equation (87) that the OHAM-based solution contains parameters $C_i$, which need to be calculated. For that purpose, one can construct the residual as follows:

$$R(v, C_i) = \mathcal{L}[\bar{s}_{OHAM}(v, C_i)] + \mathcal{N}[\bar{s}_{OHAM}(v, C_i)] + h(v), \quad i = 1, 2, \dots, s \qquad (89)$$

where $\bar{s}_{OHAM}(v, C_i)$ is the approximate solution. When $R(v, C_i) = 0$, $\bar{s}_{OHAM}(v, C_i)$ becomes the exact solution to the equation. One of the ways to obtain the optimal $C_i$ is the minimization of squared residual error, i.e.,

$$J(C_i) = \int_{v \epsilon D} R^2(v, C_i) du, \quad i = 1, 2, \dots, s \qquad (90)$$

where $D = [0.1, 10]$ is the domain of the problem. The minimization of Equation (90) leads to a system of algebraic equations as follows:

$$\frac{\partial J}{\partial C_1} = \frac{\partial J}{\partial C_2} = \dots = \frac{\partial J}{\partial C_s} = 0 \qquad (91)$$

One can obtain the optimal values of parameters by solving this system of equations numerically. Here, we used the MATLAB script *fminsearch*, which minimizes an unconstrained multivariable function. It was found that only a three-term solution produces accurate values. Therefore, a three-term OHAM solution was computed and compared

with the Theis solution, as shown in Figure 10. It can be seen that just three terms of the OHAM-based series agree well with the corresponding analytical solution to the problem. For a quantitative assessment, we also compare the numerical values of the solutions in Table 4. A flowchart containing the steps of OHAM is provided in Figure 11.

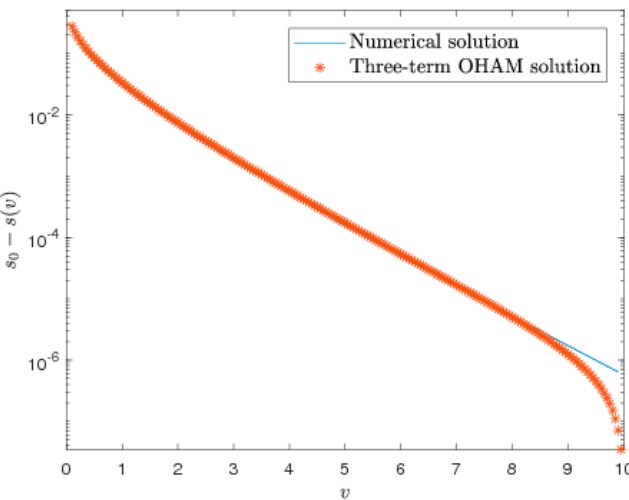

**Figure 10.** Comparison between the Theis solution and the three terms of OHAM-based solution for the selected case.

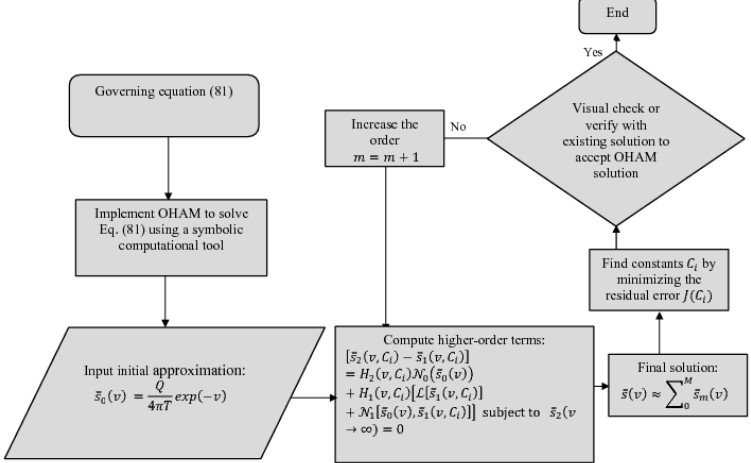

**Figure 11.** Flowchart for the OHAM solution (87).

**Table 4.** Comparison between three terms of the OHAM-based approximation and Theis solution for the selected case.

| $u$ | Numerical Solution | Three Terms of OHAM-Based Approximation |
|---|---|---|
| 0.1 | $2.523 \times 10^{-1}$ | $2.466 \times 10^{-1}$ |
| 1 | $3.036 \times 10^{-2}$ | $3.143 \times 10^{-2}$ |
| 2 | $6.768 \times 10^{-3}$ | $6.539 \times 10^{-3}$ |
| 3 | $1.806 \times 10^{-3}$ | $2.112 \times 10^{-3}$ |
| 4 | $5.230 \times 10^{-4}$ | $8.502 \times 10^{-4}$ |
| 5 | $1.589 \times 10^{-4}$ | $3.066 \times 10^{-4}$ |
| 6 | $4.983 \times 10^{-5}$ | $1.000 \times 10^{-4}$ |
| 7 | $1.598 \times 10^{-5}$ | $3.013 \times 10^{-5}$ |
| 8 | $5.213 \times 10^{-6}$ | $8.461 \times 10^{-6}$ |
| 9 | $1.723 \times 10^{-6}$ | $2.283 \times 10^{-6}$ |
| 10 | $5.753 \times 10^{-7}$ | $5.978 \times 10^{-7}$ |

### 4.5. Comparison between Different Approximations

In this section, we compare the approximations given by series approximations Equations (35), (38), (39), (42), (45), (64), (78) and (87). For that purpose, we considered the same test parameters, i.e., $Q = 4 \times 10^{-3}$ m$^3$s$^{-1}$, $T = 0.0023$ m$^2$s$^{-1}$, and $S = 7.5 \times 10^{-4}$. Moreover, for each of the cases, we computed the well function numerically using 'integral' of MATLAB to obtain the main solution. The series Equation (35) with 40 terms, Equation (38) with 10 terms, 10th-order HAM-based solution, four-term HPM solution, and three-term OHAM solution were considered. Importantly, it may be noted that the numerical values of solutions are very small, which can make the computations ill-posed or produce numerical instabilities. To that end, logarithmic form for the error was considered. Specially, we checked the performances of the approximations by calculating the percentage error as PE (%) $= 100 \times \frac{(\ln W_{num} - \ln W_{apprx})}{\ln W_{num}}$, where $W_{num}$ and $W_{apprx}$ are the values of $W(v)$ obtained from the Theis solution and the corresponding approximation, respectively. The percentage errors were calculated for the approximations and compared, as shown in Figure 12. It can be seen that among the series approximations, HAM- and OHAM-based approximations provide accurate approximations for the problem. On the other hand, the HPM-based solution is shown only within a small domain, as the solution provides accurate values there. Further, series approximation given by [12–14] are reasonably accurate within the domain. The different homotopy-based methods provide solutions that are valid within a certain range of the domain. The HAM- and OHAM-based approximations are more accurate and valid for larger domains, as they contain convergence-control parameters, which monitor the rate and radius of convergence of the series solutions. Further, the OHAM-based solution is more preferable due to its ability to provide an accurate approximation with just two–three terms of the series. Finally, it is concluded that while the homotopy-based methods do not produce as accurate solutions as do the closed-form formulae available in the literature, they are better than the series expansions and also may be improved further using different sets of base functions, linear operators, and initial approximations. It may also be noted that the proposed study differs from the existing empirical formula-based work from the viewpoint of its derivation, which starts from the governing differential equation. Therefore, the approach is flexible to use when the flow configuration is different, and the model parameters vary.

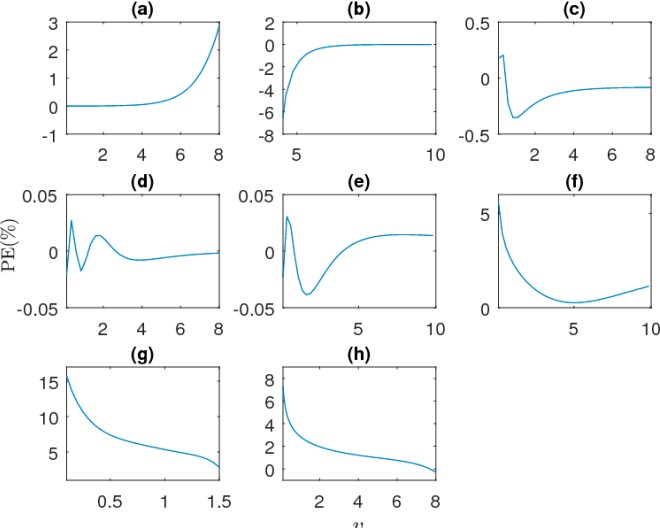

**Figure 12.** Percentage errors of the approximations: (**a**) Equation (35) with 40 terms, (**b**) Equation (38) with 10 terms, (**c**) Equation (39) [12], (**d**) Equation (42) [13], (**e**) Equation (45) [14], (**f**) Equation (64) (10th-order HAM solution), (**g**) Equation (78) (four-term HPM solution), and (**h**) Equation (87) (three-term OHAM solution).

## 5. Concluding Remarks

Theis first derived the analytical solution for the governing partial differential equation representing saturated unsteady flow in a horizontal confined aquifer. The solution was presented in the form of an exponential integral and is popularly known as the Theis solution. This exponential integral, also called the well function, requires an approximation that is simple to adapt and reliable for practical applications. Researchers have provided several approximations for the well function using series expansion, numerical approximation, etc. However, most of these approximations are valid either within a restricted domain or for a large number of terms of the series. This work directly solves the governing PDE analytically after converting it to a BVP using a similarity transformation. The HAM, HPM, and OHAM are used to obtain the solution analytically in the form of a series. It is found that both HAM and OHAM provide accurate solutions to the problem for a sufficiently large domain. Specifically, ten terms of the HAM series and three terms of the OHAM series provide the approximation well when compared with the original Theis solution. On the other hand, HPM also produces an accurate solution within a restricted domain—this is desirable, as the methodology does not contain a convergence-control parameter like HAM. Several series and numerical approximations are validated using a test case. Further, the proposed approximations are compared with the existing series and numerical approximations by calculating the percentage error. It is seen that the proposed approximations are reliable for predicting the drawdown. Because this study derives the solutions starting directly from the governing equation, the methodology adopted here can be extended to other kinds of flow configurations to find efficient solutions.

**Author Contributions:** Conceptualization, M.K. and V.P.S.; methodology, M.K.; software, M.K.; validation, M.K.; writing—original draft preparation, M.K.; writing—review and editing, V.P.S.; visualization, M.K. and V.P.S.; supervision, V.P.S. All authors have read and agreed to the published version of the manuscript.

**Funding:** This research received no external funding.

**Data Availability Statement:** No new data were created.

**Conflicts of Interest:** The authors declare no conflict of interest.

## Appendix A. Convergence Theorems

The theoretical convergence theorems of the HAM- and OHAM-based solutions Equations (64) and (87) are given here.

*Appendix A.1. Convergence Theorem of HAM-Based Solution*

The convergence theorem for the HAM-based solutions given by Equation (64) can be proved using the following theorems.

**Theorem A1.** *If the homotopy series* $\sum\limits_{m=0}^{\infty} \bar{s}_m(v)$, $\sum\limits_{m=0}^{\infty} \bar{s}_m\prime(v)$, *and* $\sum\limits_{m=0}^{\infty} \bar{s}_m''(v)$ *converge, then* $R_m(\vec{\bar{s}}_{m-1})$ *given by Equation (58) satisfies the relation* $\sum\limits_{m=1}^{\infty} R_m(\vec{\bar{s}}_{m-1}) = 0$. *[Here '$\prime$' and '$''$' denote the first and second derivatives with respect to v].*

**Proof.** The auxiliary linear operator is defined as follows:

$$\mathcal{L}[\bar{s}] = \frac{\partial^2 \bar{s}}{\partial v^2} \tag{A1}$$

According to Equation (4), we obtain:

$$\mathcal{L}[\bar{s}_1] = \hbar R_1(\vec{\bar{s}}_0) \tag{A2}$$

$$\mathcal{L}[\bar{s}_2 - \bar{s}_1] = \hbar R_2(\overrightarrow{\bar{s}_1}) \tag{A3}$$

$$\mathcal{L}[\bar{s}_3 - \bar{s}_2] = \hbar R_3(\overrightarrow{\bar{s}_2}) \tag{A4}$$

$$\mathcal{L}[\bar{s}_m - \bar{s}_{m-1}] = \hbar R_m(\overrightarrow{\bar{s}_{m-1}}) \tag{A5}$$

Adding all the above terms, we get:

$$\mathcal{L}[\bar{s}_m] = \hbar \sum_{k=1}^{m} R_k(\overrightarrow{\bar{s}_{k-1}}) \tag{A6}$$

Since the series $\sum_{m=0}^{\infty} \bar{s}_m(v)$, $\sum_{m=0}^{\infty} \bar{s}_m\prime(v)$ , and $\sum_{m=0}^{\infty} \bar{s}_m''(v)$ are convergent, we have $\lim_{m\to\infty} \bar{s}_m(v) = 0$, $\lim_{m\to\infty} \bar{s}_m\prime(v) = 0$, and $\lim_{m\to\infty} \bar{s}_m''(v) = 0$. Now, recalling the above summand and taking the limit, the required result follows as:

$$\hbar \sum_{k=1}^{\infty} R_k(\overrightarrow{\bar{s}_{k-1}}) = \hbar \lim_{m\to\infty} \sum_{k=1}^{m} R_k(\overrightarrow{\bar{s}_{k-1}}) = \lim_{m\to\infty} \mathcal{L}[\bar{s}_m] = \lim_{m\to\infty} \bar{s}_m''(v) = 0 \tag{A7}$$

$\square$

**Theorem A2.** *If $\hbar$ is properly chosen so that the series $\sum_{m=0}^{\infty} \bar{s}_m(v)$, $\sum_{m=0}^{\infty} \bar{s}_m\prime(v)$ , and $\sum_{m=0}^{\infty} \bar{s}_m''(v)$ converge absolutely to $\bar{s}(v)$, $\bar{s}\prime(v)$, and $\bar{s}''(v)$, respectively, then the homotopy series $\sum_{m=0}^{\infty} \bar{s}_m(v)$ satisfies the original governing Equation (54).*

**Proof.** Theorem A1 shows that if $\sum_{m=0}^{\infty} \bar{s}_m(v)$, $\sum_{m=0}^{\infty} \bar{s}_m\prime(v)$ , and $\sum_{m=0}^{\infty} \bar{s}_m''(v)$ converge, then $\sum_{m=1}^{\infty} R_m(\overrightarrow{\bar{s}}_{m-1}) = 0$.

Therefore, using the expression in Equation (58), we have:

$$v \sum_{m=0}^{\infty} \bar{s}_m'' + (1+v) \sum_{m=0}^{\infty} \bar{s}_m\prime = 0 \tag{A8}$$

which is basically the original governing Equation (54). Furthermore, subject to the boundary conditions $\bar{s}_0(\infty) = 0$, $\lim_{v\to 0}\left(v\frac{d\bar{s}_0}{dv}\right) = -\frac{Q}{4\pi T}$, and the conditions for the higher-order deformation equation $\bar{s}_m(\infty) = 0$, $\lim_{v\to 0}\left(v\frac{d\bar{s}_m}{dv}\right) = 0$, for $m \geq 1$, we easily obtain $\sum_{m=0}^{\infty} \bar{s}_m(\infty) = 0$ and $\lim_{v\to 0}(v \sum_{m=0}^{\infty} \bar{s}_m\prime) = -\frac{Q}{4\pi T}$. Hence, the convergence result follows. $\square$

*Appendix A.2. Convergence Theorem of OHAM-Based Solution*

**Theorem A3.** *If the series $\bar{s}_0(v) + \sum_{j=1}^{\infty} \bar{s}_j(v, C_i)$, $i = 1, 2, \ldots, s$ converges, where $\bar{s}_j(v, C_i)$ are governed by Equations (83), (85) and (86), then Equation (87) is a solution of the original Equation (81).*

**Proof.** Based on the choice of the auxiliary function, we suppose that the series Equation (25) is convergent. Then, we have:

$$\lim_{j \to \infty} \bar{s}_j(v, C_i) = 0, \quad i = 1, 2, \ldots, s \tag{A9}$$

One can write:

$$
\begin{aligned}
\bar{s}_j(v, C_i) = \bar{s}_0(v, C_i) \quad & + [\bar{s}_1(v, C_i) - \bar{s}_0(v, C_i)] \\
& + [\bar{s}_2(v, C_i) - \bar{s}_1(v, C_i)] + \ldots + [\bar{s}_j(v, C_i) - \bar{s}_{j-1}(v, C_i)] \\
& = \bar{s}_0(v, C_i) + \sum_{k=1}^{j} [\bar{s}_k(v, C_i) - \bar{s}_{k-1}(v, C_i)], \; i = 1, 2, \ldots, s
\end{aligned}
\tag{A10}
$$

Using Equation (A10), one can obtain from Equation (A9):

$$0 = \lim_{j \to \infty} \bar{s}_j(v, C_i) = \bar{s}_0(v, C_i) + \sum_{k=1}^{j} [\bar{s}_k(v, C_i) - \bar{s}_{k-1}(v, C_i)], \quad i = 1, 2, \ldots, s \tag{A11}$$

Equation (A11) can be rearranged as:

$$0 = \bar{s}_0(v, C_i) + h(v) - h(v) + [\bar{s}_1(v, C_i) - \bar{s}_0(v, C_i)] + \sum_{k=2}^{\infty} [\bar{s}_k(v, C_i) - \bar{s}_{k-1}(v, C_i)], \; i = 1, 2, \ldots, s \tag{A12}$$

Using the property of the linear operator, i.e., $\mathcal{L}[A_1(v) + A_2(v)] = \mathcal{L}[A_1(v)] + \mathcal{L}[A_2(v)]$ and $\mathcal{L}(0) = 0$, we have:

$$
\begin{aligned}
0 = \mathcal{L}(0) \quad & = \mathcal{L}[\bar{s}_0(v, C_i)] + h(v) + \mathcal{L}[\bar{s}_1(v, C_i)] - (\mathcal{L}[\bar{s}_0(v, C_i)] + h(v)) + \sum_{k=2}^{\infty} (\mathcal{L}[\bar{s}_k(v, C_i)] - \mathcal{L}[\bar{s}_{k-1}(v, C_i)]) \\
& = H_1(v, C_i) \mathcal{N}_0[\bar{s}_0(v, C_i)] \\
& \quad + \sum_{k=2}^{\infty} (H_k(v, C_i) \mathcal{N}_0[\bar{s}_0(v, C_i)] \\
& \quad + \sum_{l=1}^{k-1} H_l(v, C_i) [\mathcal{L}[\bar{s}_{k-l}(v, C_i)] + \mathcal{N}_{k-l}[\bar{s}_0(v, C_i), \bar{s}_1(v, C_i), \ldots, \bar{s}_{k-l}(v, C_i)]]) \\
& = \left[ \sum_{k=1}^{\infty} H_k(v, C_i) \right] \mathcal{N}_0[\bar{s}_0(v, C_i)] \\
& \quad + \sum_{k=2}^{\infty} \sum_{l=1}^{k-1} H_l(v, C_i) [\mathcal{L}[\bar{s}_{k-l}(v, C_i)] + \mathcal{N}_{k-l}[\bar{s}_0(v, C_i), \bar{s}_1(v, C_i), \ldots, \bar{s}_{k-l}(v, C_i)]] \\
& = H(v, C_i) \mathcal{N}_0[\bar{s}_0(v, C_i)] + \sum_{k=2}^{\infty} \sum_{l=1}^{k-1} H_l(v, C_i) [\mathcal{L}[\bar{s}_{k-l}(v, C_i)] + \mathcal{N}_{k-l}[\bar{s}_0(v, C_i), \bar{s}_1(v, C_i), \ldots, \bar{s}_{k-l}(v, C_i)]] \\
& = H(v, C_i) \mathcal{N}_0[\bar{s}_0(v, C_i)] + \sum_{k=1}^{\infty} H_k(v, C_i) \sum_{p=1}^{\infty} [\mathcal{L}[\bar{s}_p(v, C_i)] + \mathcal{N}_p[\bar{s}_0(v, C_i), \bar{s}_1(v, C_i), \ldots, \bar{s}_p(v, C_i)]] \\
& = H(v, C_i) \mathcal{N}_0[\bar{s}_0(v, C_i)] + H(v, C_i)[\mathcal{L}(\sum_{p=1}^{\infty} \bar{s}_p(v, C_i)) + \sum_{p=1}^{\infty} \mathcal{N}_p[\bar{s}_0(v, C_i), \bar{s}_1(v, C_i), \ldots, \bar{s}_p(v, C_i)]] \\
& = H(v, C_i) \mathcal{N}_0[\bar{s}_0(v, C_i)] + H(v, C_i)[\mathcal{L}(\bar{s}(v, C_i)) - \mathcal{L}(\bar{s}_0(v, C_i)) + \mathcal{N}(\bar{s}(v, C_i)) - \mathcal{N}(\bar{s}_0(v, C_i))] \\
& = H(v, C_i) \mathcal{N}_0[\bar{s}_0(v, C_i)] \\
& \quad + H(v, C_i)[\mathcal{L}(\bar{s}(v, C_i)) - [\mathcal{L}(\bar{s}_0(v, C_i)) + h(v)] + h(v) + \mathcal{N}(\bar{s}(v, C_i)) - \mathcal{N}(\bar{s}_0(v, C_i))] \\
& = H(v, C_i)[\mathcal{L}(\bar{s}(v, C_i)) + h(v) + \mathcal{N}(\bar{s}(v, C_i))], \quad i = 1, 2, \ldots, s
\end{aligned}
\tag{A13}
$$

Now, since $H(v, C_i) \neq 0$, from Equation (A13), we have

$$\mathcal{L}(\bar{s}(v, C_i)) + h(v) + \mathcal{N}(\bar{s}(v, C_i)) = 0, \quad i = 1, 2, \ldots, s \tag{A14}$$

which shows that $\bar{s}(v, C_i)$ is the exact solution of Equation (81). $\square$

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
