# Peer review of "Analytical Approximations of Well Function by Solving the Governing Differential Equation Representing Unsteady Groundwater Flow in a Confined Aquifer"

_mathematics, doi:10.3390/math11071652_

Round 1

Reviewer 1 Report

The manuscript "Analytical Approximations of Well Function by Solving the Governing Differential Equation Representing Unsteady Groundwater Flow in a Confined Aquifer" is well-written by the authors and the work is very much useful. I accept the manuscript for possible publication. However, my only concern is that the "Conclusions are not supported by the results". With that change, the paper can be accepted. 

As such there are no specific queries as the manuscript is well written.  I only found that the conclusions are not given with results. Further, as the results are discussed purely on mathematical approximations of considering a specific number of terms in the series, the results behave accordingly.  However, following few specific queries can be added to my review: 

1) Line 612-613 and Figure 4, Equation 38 fail to perform for v<4, what could be the reason? 

2) What is the limitation of present work on the value of v (that is lower and upper limits). Because when we compare Figures 6 with HAM (0<v<10), Figure 8 for HPM (0<v<2) and Figure 10 for OHAM (0<v<2), which one would be most suitable for the common v range. The additional discussion can be added in the manuscript. 

Author Response

Reply to Reviewer #1:

The authors express their sincere thanks to Reviewer #1 for a critical review and comments on the manuscript. For each comment in the review report, we provide a detailed reply and mention the necessary changes made in the revised manuscript. These comments remarkably helped improve the previously submitted version of this manuscript.

All the changes are highlighted.

1. The manuscript "Analytical Approximations of Well Function by Solving the Governing Differential Equation Representing Unsteady Groundwater Flow in a Confined Aquifer" is well-written by the authors and the work is very much useful. I accept the manuscript for possible publication. However, my only concern is that the "Conclusions are not supported by the results". With that change, the paper can be accepted.

Reply: Thanks for your encouraging comments. In the revised version, we have rewritten the conclusion part incorporating the main outcomes of the work.

 2. As such there are no specific queries as the manuscript is well written. I only found that the conclusions are not given with results. Further, as the results are discussed purely on mathematical approximations of considering a specific number of terms in the series, the results behave accordingly. 

Reply: Thanks for your comment.

3. Line 612-613 and Figure 4, Equation 38 fail to perform for v<4, what could be the reason?

Reply: Thanks for the comment. Eq. (38) is an asymptotic expansion, which is valid for the large values of v. These are discussed in lines: 355-367.

We can see from Eq. (38) that the remaining term (term involving Big-O notation) becomes larger when v is small – this is why the series does not work for small v-values.

4. What is the limitation of present work on the value of v (that is lower and upper limits). Because when we compare Figures 6 with HAM (0<v<10), Figure 8 for HPM (0<v<2) and Figure 10 for OHAM (0<v<2), which one would be most suitable for the common v range. The additional discussion can be added in the manuscript.

Reply: Thanks for the useful comment. We have revised it. Please see lines: 765-776.

Reviewer 2 Report

Manuscript ID: mathematics-2248286

Title: Analytical Approximations of Well Function by Solving the Governing Differential 2 Equation Representing Unsteady Groundwater Flow in a Confined Aquifer

The following comments needs to be addressed:

1.     The contribution of the study to existing literature and novelty should be emphasized end of the introduction section.

2.     Assumptions made for the model need to be stated clearly.

3.     The physical phenomena behind the presented study need to be mentioned in the paper.

4.     In mathematical formulation equation numbers are not in proper order.

5.     Add the colour figures and improve the quality of figures.

6.     It is recommended the authors try to explain the novelty of the paper as clear as possible and explain the research gap they are trying to fill.

7.     The motivation and objective should be improved.

8.     The introduction section is very weak, please improve the introduction section.

9.     Findings obtained from the study can be evaluated more comprehensively in terms of reason-result relationship in order to have widespread effect.

10.  The results and discussion section should be improve.

Author Response

Reply to Reviewer #2:

The authors express their sincere thanks to Reviewer #2 for a critical review and comments on the manuscript. For each comment in the review report, we provide a detailed reply and mention the necessary changes made in the revised manuscript. These comments remarkably helped improve the previously submitted version of this manuscript.

All the changes are highlighted.

1. The contribution of the study to existing literature and novelty should be emphasized end of the introduction section.

 Reply: Thanks for the useful comment. We have emphasized them in the revised version. Please see lines: 86-95.

 2. Assumptions made for the model need to be stated clearly.

Reply: Thank you. The assumptions made for the derivation of the model are given in lines 268-274. Please see.

3. The physical phenomena behind the presented study need to be mentioned in the paper.

Reply: Thank you. They are given in lines 268-274. Please see.

4. In mathematical formulation equation numbers are not in proper order.

Reply: Thanks. We have rechecked the equation numbers. We think they are in proper order. Please see.

5. Add the colour figures and improve the quality of figures.

Reply: Thanks. We have used colour figures wherever appropriate. Also, the manuscript is written in Microsoft word, maybe because of that the figures’ quality seem low. However, we will upload the figures separately.

6. It is recommended the authors try to explain the novelty of the paper as clear as possible and explain the research gap they are trying to fill.

Reply: Thank you. It is done in the ‘Introduction’ section. Please see lines: 86-95.

7. The motivation and objective should be improved.

Reply: Thank you. Please see lines: 86-95.

8. The introduction section is very weak, please improve the introduction section.

Reply: Thank you. The work is based on the approximation of well function (exponential integral). Therefore, the ‘Introduction’ section is written within the context of this work. However, we have modified the motivation and objective’s part. Please see lines 86-95.

9. Findings obtained from the study can be evaluated more comprehensively in terms of reason-result relationship in order to have widespread effect.

Reply: Thank you. It is done, specifically for the comparison of different approximations given by Fig. 12. Please see lines: 765-776.

10. The results and discussion section should be improved.

Reply: Thank you. Please see lines 765-776.

Reviewer 3 Report

Although this research is not entirely new, it is good and will be very helpful to aspiring researchers. The writing in the manuscript is admirable and orderly. I have a few little recommendations, which are as follows:

1)      The authors used the three methods such as HAM-Based Solution, HPM-Based Solution and OHAM-Based Solution. All three methods are well represented and explained.

2)      The authors may change 10-, 20-, and 40- terms by 10, 20, and 40 terms at page no. 21 in third line from bottom, and do the same correction where it required.

3)      In figure 3, 2-, 5- , 100- term approximation replace by 2, 5, and 100 term in figure caption.

4)      The sentence is started To that end, in line 276. It is not looking correct, sentence should be modified. Please do the same correction where it is needed.

5)      Ei should be replace by Ei in equations 36 and 37 and do the same correction where it is required.

6)      Figures 2, 3, 4 and 6 are mentioned unit less. Please mentioned the unit of v and w(v) in figs. 2, 3, 4 and 6.

7)      Authors should give the latest references related to their work in reference section.   

8)      Remove the typographical errors.

Author Response

Reply to Reviewer #3:

The authors express their sincere thanks to Reviewer #3 for a critical review and comments on the manuscript. For each comment in the review report, we provide a detailed reply and mention the necessary changes made in the revised manuscript. These comments remarkably helped improve the previously submitted version of this manuscript.

All the changes are highlighted.

1. Although this research is not entirely new, it is good and will be very helpful to aspiring researchers. The writing in the manuscript is admirable and orderly.

 Reply: Thanks for the encouraging comments.

2. The authors used the three methods such as HAM-Based Solution, HPM-Based Solution and OHAM-Based Solution. All three methods are well represented and explained.

Reply: Thanks for the comment.

3. The authors may change 10-, 20-, and 40- terms by 10, 20, and 40 terms at page no. 21 in third line from bottom, and do the same correction where it required.

 Reply: Thanks for the useful comment. We have done it throughout the manuscript. Please see the highlighted parts.

 4. In figure 3, 2-, 5- , 100- term approximation replace by 2, 5, and 100 term in figure caption.

 Reply: Thanks for the useful comment. We have done it throughout the manuscript. Please see the highlighted parts.

 5. The sentence is started To that end, in line 276. It is not looking correct, sentence should be modified. Please do the same correction where it is needed.

 Reply: Thanks. We have replaced it with ‘Therefore’. Please see. Also, we have corrected it wherever needed.

 6. Ei should be replaced by Ei in equations 36 and 37 and do the same correction where it is required.

 Reply: Thanks. Actually, Ei is the standard notation for the exponential integral. All the books use the same notation for this. It is may be because denoting it using ‘i’ as a subscript does not carry any specific meaning.

 7. Figures 2, 3, 4 and 6 are mentioned unit less. Please mentioned the unit of v and w(v) in figs. 2, 3, 4 and 6.

 Reply: Thank you. The variable v is defined (just after eq. 31) in terms of the temporal and radial coordinate, transmittivity, and storativity. While each of them has units, the variable v is dimensionless. Accordingly, W(v) is also dimensionless.

 8. Authors should give the latest references related to their work in reference section.

 Reply: As mentioned in previous replies and also in the ‘Introduction’ section, the specific objective of this paper is to provide approximations for the well function (exponential integral) using homotopy-based methods. Accordingly, the specific literature is cited. For example, the most recent one proposed by Vatankhah (2014) is added and compared with the present approximations.

 9. Remove the typographical errors.

 Reply: Thank you. We have thoroughly checked the revised version and made necessary changes wherever needed.

Reviewer 4 Report

In this article, the authors have examined Homotopy based methods.

  According to the contents presented in this article, it can be read as a descriptive article after a general review.

  Therefore, in my opinion, this article does not have the standard of a research article.

Author Response

Reply to Reviewer #4:

The authors express their sincere thanks to Reviewer #4 for a critical review and comments on the manuscript. For each comment in the review report, we provide a detailed reply and mention the necessary changes made in the revised manuscript. These comments remarkably helped improve the previously submitted version of this manuscript.

All the changes are highlighted.

1. In this article, the authors have examined Homotopy based methods. According to the contents presented in this article, it can be read as a descriptive article after a general review. Therefore, in my opinion, this article does not have the standard of a research article.

 Reply: Thanks for your review. Following the other reviewers’ comments, we have rewritten the motivation and objectives of the work in the ‘Introduction’ section. Also, some parts are improved. Please see the revised manuscript.

Round 2

Reviewer 4 Report

In Tables 2 and 3, in the first column, which numerical method is meant, which method was used for the exchange.

Author Response

Comment: In Tables 2 and 3, in the first column, which numerical method is meant, which method was used for the exchange.

Reply: Thanks for the useful comment. The numerical solution is obtained for the integral involved in Theis solution (Eq. 31 or 32). We have used the MATLAB script ‘integral’, which uses the global adaptive quadrature rule (Shampine 2008). 

We have added some discussion in lines 649-652, which are highlighted. Please see the revised version.